# Perceived Benefits and Costs of Owning a Pet in a Megapolis: An Ecosystem Services Perspective

Anastasia Konstantinova [1,*] , Victor Matasov [1] , Anna Filyushkina [2] and Viacheslav Vasenev [1,3]

1   Department of Landscape Design and Sustainable Ecosystems, Peoples' Friendship University of Russia—RUDN University, 117198 Moscow, Russia; matasov-vm@rudn.ru (V.M.); slava.vasenev@wur.nl (V.V.)
2   Institute for Environmental Studies, Vrije Universiteit Amsterdam, 1081 HV Amsterdam, The Netherlands; anna.filyushkina@vu.nl
3   Department of Soil Geography and Landscape, Wageningen University and Research Centre (WUR), 6708 PB Wageningen, The Netherlands
*   Correspondence: konstantinova-av@rudn.ru

**Abstract:** Ongoing urbanization has led to a significant increase in the number of pets and has altered the relationships between pets and owners from primarily utilitarian to cultural (e.g., entertainment and health improvement). Existing classifications of ecosystem services (ES) (e.g., CICES) and nature's contributions to people (NCP) explicitly consider only the ES provided by livestock and wild animals. This study attempted to translate perceived benefits and costs from owning pets (dogs or cats) in a megapolis into ES and disservices frameworks. The data were collected via an online questionnaire distributed through social media among residents of Moscow (Russia). The study showed that pets contribute to the well-being of city dwellers, for which owners are willing to put up with some potential risks and also bear monetary costs. Reasons for owning a pet have been translated into ES and NCPs ranging from regulating (4%) to provisioning (1%). However, cultural services linked to mental (26%) and physical (32%) health, spiritual, symbolic interaction (19%), and educational values (16%) have been the most prominent group. Considering an increase in pet owners, the interests and needs of this distinct stakeholder group need to be taken into account in urban planning and management. Pets' integration into classifications and thus assessments of the urban ES can be a crucial step towards achieving this goal.

**Keywords:** domestic animals; pets; urban planning; CICES; NCP; stakeholders; disservices





## 1. Introduction

Companion animals, or pets, are formally defined as animals we live with and that have no obvious economic and practical function, but the value we attribute to them comes from the benefits of the relationship we have with them [1]. The relationship between people and domestic animals in contemporary societies and ecosystems is a longstanding and enduring issue [2,3]. There are a number of theories that dogs were domesticated to help hunting, and to protect humans, their homes and livestock [4,5]. Cats were valued at all times as low-maintenance predators-in-residence to protect food stores and warehouses from rodents [3,6]. This appreciation played a significant role in their global spread, as they were used as pest controllers on commercial vessels [7]. The human–animal bonds and benefits people receive from their pets are also changing with time. In Asia (mostly in Cambodia, China, Thailand, and Vietnam), where approximately 15 million dogs and 4 million cats were eaten each year [8], pets are now protected by government regulations [9]. About 90% of pet owners in western countries consider their companion animals as family members [10]. Recent studies of socio-psychological benefits derived from pet ownership show that dogs act as social lubricants by encouraging strangers to meet and talk by providing a neutral topic of conversation [1,11]. It has been

repeatedly shown that stroking or petting a cat causes a short-term drop in blood pressure and/or heart rate, which is a stress moderator [11,12]. Dog-owners appear to gain more long-term health advantages than owners of cats due to more physical activity, especially walking [13–15]. Assistance animals are trained to perform different tasks for individuals with a variety of disabilities [11,12]. With growing rates of consumption and decreasing human reproduction rates, pets become the emotional surrogates of young children, which cheaper and less demanding to raise than kids, but with potential benefits to one's social life in urban communities [9]. Recent articles have shown how pets affect people's mental health in the COVID-19 pandemic [16–18].

People not only benefit from cats and dogs—this relationship also carries risks for both sides. Humans could cause harm to pets, while living with animals increases the chances of humans contracting zoonotic pathogens [19,20]. Some studies assume that city residents' higher rates of schizophrenia can be partly explained by exposure to Toxoplasma gondii oocysts excreted by cats [21,22]. On the other hand, pet ownership could result in substantial changes in urban ecosystems. Several ecological studies have quantified the scale and impact of cat or dog predation on wildlife [23–25]. Abandoned domestic cats and dogs become dangerous as generalist obligate predators, and thus when their preferred prey species decline, they can switch to other prey species or find food near settlements, so their population densities are maintained [24,26]. All these risks and problems are widely discussed in terms of regulation policies (e.g., microchipping, desexing, etc.) and management practices [27–30].

Today, in the highly urbanized world, the number of cats and dogs is growing, and their popularity as pets follows population growth in many countries [31]. The significant increase in households that keep pets is associated with the development of the pet-food industry, and, as a result, the financial costs are rising rapidly [32]. In Canada, where around 8.5 million cats and 6 million dogs are kept, the average annual cost per animal is around CAD 1000–1500. The amount Canadians spent on pets in 2018 exceeded what they spent on their hobbies, toys, and games (CAD 5.8 billion), while in the US, about USD 41 billion was already being spent on pets a decade ago [33,34]. Russia is one of the countries with the largest percentage of families who own pets [35]. According to the data published in 2019 by the Russian public opinion research center, about 68 percent of Russian families have domestic animals, mainly cats and dogs [36]. The most common pets are cats; 17.8 million of them live in families, which is the third in the world after the US and China (74.1 million and 53.1 million cats, respectively). The number of dogs in Russia is 12.5 million. This is the fourth highest after the US, Brazil and China (69.9 million, 35.8 million, 27.4 million dogs, respectively) [35]. Specifically, 9% of Russian cats and 7% of Russian dogs live in Moscow and St. Petersburg [37].

In the urban environment, pets influence peoples' way of life and well-being. As natural biotic elements they are essential parts of urban ecosystems. Their increasing numbers as well as the willingness of pet owners to endure related costs are indisputable, and have resulted in a large market of goods and services. Thus, pet owners present a distinct stakeholder group, whose interests and values ought to be considered in both ES assessments and urban planning in order to truly investigate and embrace a plurality of values. However, in both the Common International Classification of Ecosystem Services (CICES) as well as in the nature's contributions to people (NCP) classification, pets such as cats and dogs are not explicitly considered, whereas both frameworks allow for depicting the benefits derived from domestic animals such as livestock, and are responsible for provisioning and regulating services [38,39].

This study aims at contributing to this knowledge gap. To do so, first, we investigate reasons, benefits and costs of owning a pet (dog or cat) in a megapolis, as well as the availability of infrastructure as perceived by respondents using a sample of Moscow residents who are pet owners. Additionally, we also test whether there are differences between the responses of dog and cat owners. Then, we attempt to translate these benefits and costs into ES and disservice categories using CICES and NCP classifications.

## 2. Methods and Materials

### 2.1. Data Collection and Survey Distribution

In this study, Moscow was chosen as the case of a large metropolis with 12.7 million inhabitants. The survey was conducted by an online questionnaire using survey system Survio (survio.com) and Russian social network Vkontakte (vk.com) from 18 March till 20 April 2020. The questionnaire aimed to understand the preferences of pet owners in Moscow. To recruit participants, a targeted advertisement was created on the Vkontakte social network. As the target audience should include pet owners, it was formed based on people's participation in thematic groups. There were groups dedicated to selling/buying/donating pets and finding lost pets, groups of veterinary clinics and animal shelters, as well as groups with entertainment content about domestic pets. The groups were selected based on their spatial location, so that residents of different parts of the city were represented in the survey. This resulted in a total of 50 groups being selected for exposure to the advertisement with a target audience of 63,000 people. The advertising views were limited to 1 per day per person, and the budget for the entire period was USD 120.

### 2.2. Study Sampling

During the survey period there were 36,004 views of the advertisement (Table 1). The introduction page on Survio was visited by 569 people and 346 of them completed the survey. After cleaning, the final sample providing data for the present study included 229 respondents.

**Table 1.** Inclusion and exclusion criteria of the survey.

| Number of Views of the Advertisement in Vk | Number of Visits to the Entry Page on Survio | Number of Visitors Who Have Completed the Survey | Number of Moscow Residents | Number of Moscow Residents—Pet Owners | Number of Moscow Residents—Dog or (and) Cat Owners |
|---|---|---|---|---|---|
| 36,004 | 569 | 346 | 260 | 242 | 229 |

### 2.3. Questionnaire Design

The questionnaire was composed of three parts (17 questions). The first part of the questionnaire comprised 2 filter questions regarding residency in Moscow and pet owning: 1 one-choice question on number of pets, and 1 multi-choice question on type of pet. Main reasons for owning a pet and key disadvantages of pet owning were derived from the scholarly literature [40–42] and presented as multi-choice questions. Questions on the amounts (rubles per year) respondents spend on their pets were open-ended. The responses were asked to provide cost estimates separately for the following categories: food, litter, health supplies (medications, vaccinations, vitamins), beauty and grooming (bathing, trimming, styling, etc.) and other goods (clothes, toys, dishes, etc.).

The second part of the questionnaire was composed of 2 questions to reveal the preferences on walking areas in the neighborhood (multi-choice question) [43,44] and satisfaction with green areas (parks, squares, etc.) for walking with animals in the neighborhood (ranking question). Respondents were asked to assess the availability of green infrastructure as a key factor influencing pet ownership comfort in a megacity. We used a 5-point scale, according to which respondents were asked to assess to what extent they thought there were enough green areas in their area (from 1—completely insufficient to 5—completely enough).

The third part of the questionnaire comprised five socio-demographic questions covering gender, age, education level, relationship status and residence in the neighborhood. Questionnaire responses were anonymous and confidential. When cats and dogs lived in the same home, respondents were assigned to the cat and dog groups simultaneously.

### 2.4. Data Analysis

A descriptive analysis of the questionnaire data was conducted using the SPSS software (IBM SPSS v24). This analysis included data categorization and transformation and descriptive statistics. To determine if there were significant differences in responses between cat and dog owners we ran Pearson chi-square tests for categorical data as well as nonparametric Mann–Whitney U tests for ordinal data.

## 3. Results

### 3.1. Respondents Characteristics and Pet Ownership Structure

The demographic characteristics of the survey respondents are presented in Table 2. The sample was dominated by women (90.0% vs. 10.0% men). Based on the data, most pets live in families: more than half of respondents reside with children or parents (64.2%) or with a spouse (22.7%). The majority of our sample is in the age groups from 20 to 40 years old (39.7% from 20 to 30 years old, 22.7% from 30 to 40 years old). The education level of pet owners was dominated by higher education (58.5%) or an incomplete level of higher education (12.2%), while 17.5% of respondents had only completed secondary education, and 11.8% have or do not have a high school diploma.

**Table 2.** Demographic characteristic of respondents (229 pet owners residing in Moscow).

| Characteristic | Share of Respondents |
|---|---|
| Gender | 10%—Male<br>90%—Female |
| Age | 13.5%—Less than 20 years old<br>39.7%—21–30 years old<br>22.7%—31–40 years old<br>13.5%—41–50 years old<br>8.7%—51–60 years old<br>1.7%—More than 60 years old |
| Highest obtained level of education | 2.2%—Primary general (4 grades of school)<br>7.0%—Basic general (9 grades of school)<br>2.6%—Secondary general (11 grades of school)<br>17.5%—Secondary vocational<br>12.2%—Incomplete higher<br>51.1%—Higher (bachelor, specialty, master)<br>7.4%—Higher (postgraduate or doctoral studies, PhD degree) |
| Living/family circumstances | 7.9%—Alone<br>22.7%—With spouse<br>64.2%—With family (children, parents)<br>4.8%—With friend/friends<br>0.4%—no answer |

Of the 229 survey respondents, 47.6% own only a cat (cats), 24.0% own only a dog (dogs) and 28.4% own a cat (cats) and a dog (dogs) (Figure 1a). The most common pets are cats (60.1%): 38.7% outbred cats and 21.4% purebred cats (Figure 1b). In the case of dogs, 27.6% of respondents own purebred dogs and 12.4% mongrel dogs. The majority of cat owners have one cat (57.5%), while 23.6% have two cats and 18.9% have three or more cats (Figure 1c). For dogs, the situation is almost the same, but the number of respondents with one dog is highest (71.7%); those with two dogs represent 17.5% of cases and those with three or more dogs 10.8% of cases.

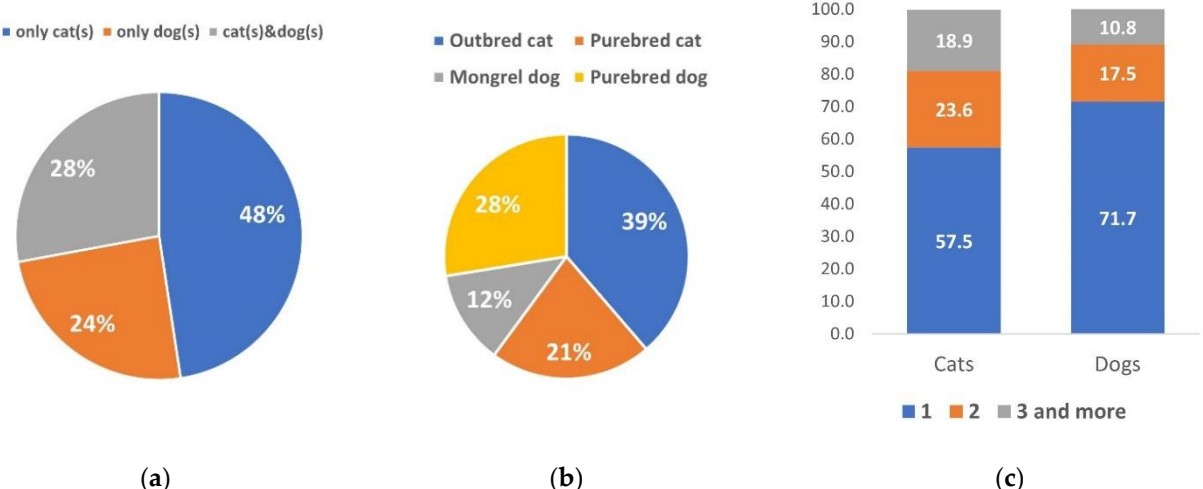

**Figure 1.** Pet ownership among respondents: (**a**) cat and dog owners ratio; (**b**) outbred and purebred cats and dogs ratio; (**c**) number ratio of cats and dogs per owner (in %).

Cats are the most popular pets, regardless of their owner's status. At the same time, it should be noted that the majority (66.7%) of single people choose cats, besides 46.2% of married couples and 46.9% of families. Besides this, people living with a friend or girlfriend are more likely to own a dog—54.5%—than a cat (36.4%).

*3.2. Reported Reasons for Owning a Pet Translated into ES and NCP*

The study revealed different reasons for owning pets, which we then translated into the ES and NCP frameworks (Table 3, Table S1 in Supplementary). For reasons that were not mentioned explicitly in either of the categorizations, we have provided a suggestion of where we believe it fits best based on a wider interpretation.

**Table 3.** Reported reasons for owning a pet translated into ES and NCP with percentages of respondents who gave them.

| Reason for Owning a Pet Reported by Respondents | Suggested Translation into ES/NCP | | | Percentage of Respondents Owning a Pet Reporting Such Reasons | |
|---|---|---|---|---|---|
| | Suggested Category and Reasoning | Equivalent in CICES [1] (If Exists) | Equivalent in NCP [2] (If Exists) | Cats | Dogs |
| For joy | Mental health as part of physical and psychological experiences | 3.1.1.1 | (16) | 26.2 | 28.4 |
| For comfort, coziness, sense of home | Sense of place as part of supporting identities | 3.2.1.1 | (17) | 21.7 | 16.7 |
| For caring/feeling responsible | Educational values as part of learning and inspiration | 3.1.2.2 | (15) | 14.3 | 16.4 |
| To shelter | Mental health as part of physical and psychological experience | 3.1.1.2 | (16) | 12.8 | 5.0 |
| Not to feel alone | Mental health as part of physical and psychological experience | 3.1.1.2 | (16) | 10.5 | 9.1 |
| For beauty/to admire | Aesthetics, both as a source of learning and inspiration and as part of physical and psychological experience | 3.1.1.2 | (15) and (16) | 7.2 | 7.6 |

**Table 3.** *Cont.*

| Reason for Owning a Pet Reported by Respondents | Suggested Translation into ES/NCP | | | Percentage of Respondents Owning a Pet Reporting Such Reasons | |
|---|---|---|---|---|---|
| | Suggested Category and Reasoning | Equivalent in CICES [1] (If Exists) | Equivalent in NCP [2] (If Exists) | Cats | Dogs |
| For health | Physical and mental health as part of physical and psychological experience | 3.1.1.1 | (16) | 2.5 | 5.6 |
| For breeding/sale | Materials and assistance and/or sense of identity and belonging as part of supporting identities or even maintenance of options | 1.2.2.1 1.2.2.2 | (13), (17) and (18) | 1.1 | 0.9 |
| To participate in exhibitions | Aesthetic enjoyment based on close contact with the pet (nature) and/or sense of belonging, connectedness to the pet and/or community with its rituals and customs | 3.1.1.1 | (16) and (17) | 0.2 | 1.2 |
| To guard | Regulation of organisms detrimental to human well-being | 2.2.3.X [3] | (10) | - | 6.4 |
| For hunting | Food and feed/provisioning contribution and/or sense of belonging, identity to the community with its rituals/relaxation and enjoyment of nature | 3.1.1.1 1.1.6.1 | (12), (16) and (17) | - | 0.3 |
| To catch mice | Pest control/regulation of organisms detrimental to humans | 2.2.3.1 | (10) | 1.3 | - |
| Other | - | - | - | 2.2 | 2.4 |

[1] Based on CICES v5.1. [2] Based on update on the classification of NCP by the Intergovernmental Science-Policy Platform on Biodiversity and Ecosystem Services. IPBES 5/INF/24. 2017. [3] X means a new class in CICES v 5.1. needs to be created.

We found that pets are primarily chosen for mental health and cultural reasons (such as sense of belonging, identity, place) and less often for money or practical/material benefits (Table 3). Several examples reflecting the category of health as part of the phycological experiences have been named, including the common answers among the respondents for cats (26.2%) and dogs (28.4%) related to joy and pleasure and not feeling alone, which was the more popular response in the case of outbred pets (mongrel dogs—12.4% and outbred cats—10.5%). Overall, we found a significant correlation between pet type and joy as the reported reason for ownership, with dogs being more frequently chosen (Pearson chi-square = 0.116; $p$ = 0.047). Regarding benefits to physical health, for dogs, again this reason was named more often (Pearson chi-square = 0.154; $p$ = 0.008). To have a dog, even a small one, means to go outside a minimum of two times per day, to walk a minimum of 30 min and contact nature, and green spaces are considered a significant benefit for personal health by respondents. The second most represented category of benefits was linked to supported identity. As a stand-alone category, it was reflected in such reasons as comfort, coziness and sense of home, which were more prominent for cat owners (21.7%; 20.4%—outbred cats and 22.8%—purebred cats) and were only reported by 16.7% of dog owners (15.9%—mongrel dogs and 17.2%—purebred dogs). Another well-represented category was linked to education, and learning and associated with responsibility and care. More than 27.1% of cat owners (for cats, this reason was named more often, with a

significance level of 0.01 (Pearson chi-square= −0.205; $p$ = 0.000)) and 21.4% of dog owners decided to take a pet from the shelter, care for and feel responsible themselves, or educate their children in this skill. The aesthetic reasons, provided by 7.2% of respondents with cats and 7.6% respondents with dogs, could be connected to several ES/NCP categories such as learning and education and physiological health, based on the exact value and experience the respondent refers to.

In contrast to cultural and mental reasons, the answers associated with pets' historical and practical/materials functions have been less commonly named by respondents. However, these functions (guarding (6.4%), hunting (0.3%) and catching mice (1.3%)) in the present day could also relate to a combination of ES/NCP, depending on the exact value the respondent places on them—anything from material or provisioning and a sense of belonging in the case of hunting, to regulating in the case of guarding or catching mice. As expected, catching mice was named significantly more often for cats (Pearson chi-square = −0.120; $p$ = 0.040), whereas hunting and guarding was named more often for dogs (Pearson chi-square = 0.225; $p$ = 0.000 and Pearson chi-square = 0.325; $p$ = 0.000, respectively). Breeding and participating in exhibitions were among the less prominent reasons for owning a pet in our sample. They are not explicitly mentioned in any of the ES or NCP categories. Since they could be based on several different motives, we found them linked to different categories, such as supporting a sense of belonging and identity (for example, being an owner of a specific breed), the aesthetic enjoyment of being close to different breeds and, finally, the material benefits of breeding a new generation.

We found that more than 90% of ES provided by pets falls under the overall category of cultural services (Figure 2). Pets are primarily chosen for physical (32%) and mental (26%) health support, as well as for indirect spiritual, symbolic interaction (19%) and direct intellectual interaction for educational purposes (16%). Services connected to practical and material benefits, such as regulating services (controlling pests and invasive species/regulation of organisms detrimental to human well-being/using for replenishing stock or for breeding) or provisioning, were less prevalent in the respondents' answers (7% in total).

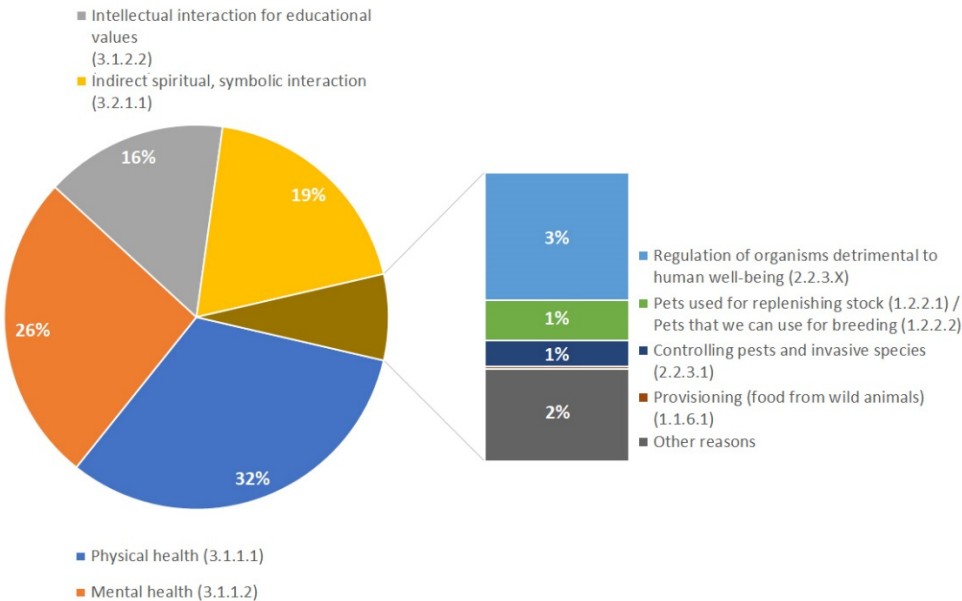

**Figure 2.** Breakdown of reasons reported by participants for owning a pet translated and grouped into categories of ES using CICES classification.

### 3.3. Disservices of Owning a Pet Translated from Perceived Disadvantages

The disservices related to owning a pet in a megapolis have been derived from respondents by asking them about the perceived/experienced disadvantages. Since there

is no structured categorization of disservices as with services, these are presented as they were named by participants (Figure 3 and Table S1 in Supplementary). The most commonly named disadvantages are dirt and wool all over the house (39.0% of dog owners and 37.6% of cat owners). Other significant disadvantages include damaged property (furniture, wallpaper, etc.) and noise, which were named by 30.5% and 16.1% of cat owners and 27.7% and 16.4% of dog owners, respectively. Problems connected with health, such as allergies (6.7% of dogs and 5.7% of cats) and contagious diseases (0% of dogs and 0.7% of cats) were the least reported in our sample. The rest of the shortcomings identified in the "other disadvantages" category reflect pets' character, problems with their health and the need to wake up early in the morning, but more often, respondents wrote that there were no problems. There is no significant correlation between pet type and the following perceived disadvantages: noise (Pearson chi-square = 0.006; $p$ = 0.916), dirt and wool all over the house (Pearson chi-square = 0.016; $p$ = 0.779), damaged property (Pearson chi-square = −0.043; $p$ = 0.462), allergy (Pearson chi-square = 0.009; $p$ = 0.872), contagious diseases (Pearson Chi-Square = −0.069; $p$ = 0.240).

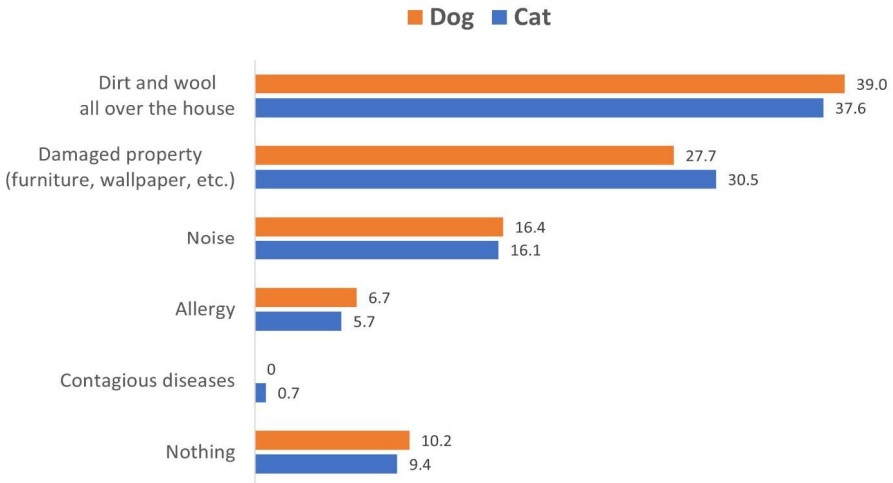

**Figure 3.** Disadvantages associated with owning a pet reported by respondents (%).

### 3.4. Costs of Owning a Pet as Reported by Respondents

Owning a pet brings not only benefits and limitations, but also involves financial costs. Our findings show that, as expected, there is a significant difference between the costs of owning a dog versus those for a cat ($p$ = 0.000; U = 6393.000; Z = −3.749), with the former being more expensive (Figure 4 and Table S2 in Supplementary). On average, these costs are 1.7 times higher for the food and health supplies, including medications, vaccinations and vitamins, 2.8 times higher for different goods, and 11.8 times higher for maintaining beauty (bathing, trimming, styling).

However, the costs breakdown (Figure 5) is similar for cats and dogs. From 60% to 65.8% of expenses are spent on food. The most considerable expense under this item is on mongrel dogs (65.8%) and mongrel cats (64.1%). Medical expenses range from 12.7% to 18.6%, and the highest percentage is for mongrel dogs. Further, in the case of cats, there are the expenses for cat litter—16.8% of the total. The owners of purebred dogs devote 10.4% to beauty and grooming, while the owners of mongrel dogs devote 8.2%, purebred cats 2.1%, and purebred cats 1.4% to this end. Other goods such as clothes, toys and dishes are also more expensive for purebred dog owners (13.3%). The owners of small pedigree dogs may include the cost of a dog toilet in this part of the expenses, as the question of spending on cat litter was not included in the questionnaire for dog owners. However, it is typical in Russia that even small dogs are walked on the street, and, unfortunately, the practice of collecting excrement is just beginning to form. Therefore, uncovered soil is seen as a free toilet for dogs.

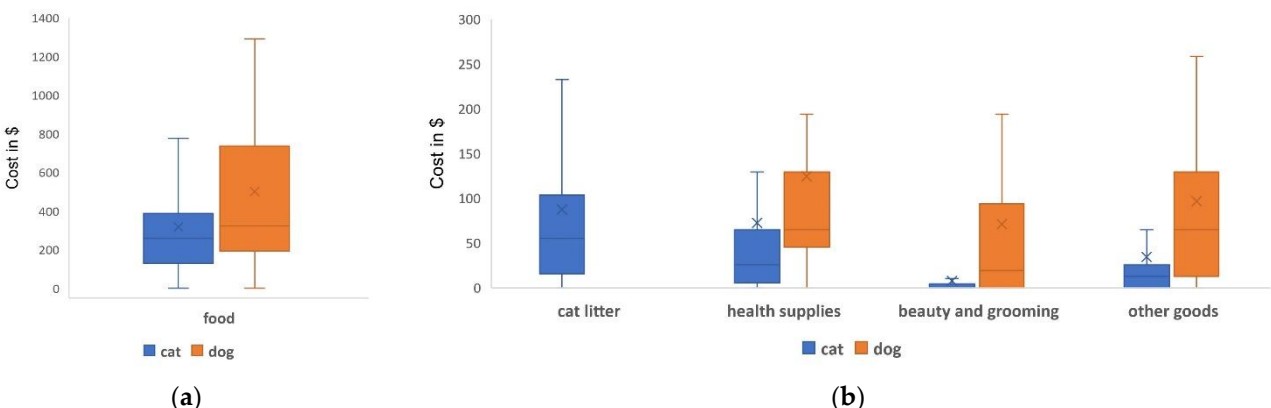

**Figure 4.** The costs of pet ownership (minimum and maximum values, mean and median, quartiles 1 and 3) in USD per year: (**a**) food cost and (**b**) cat litter, health supplies, beauty and grooming cost and other goods.

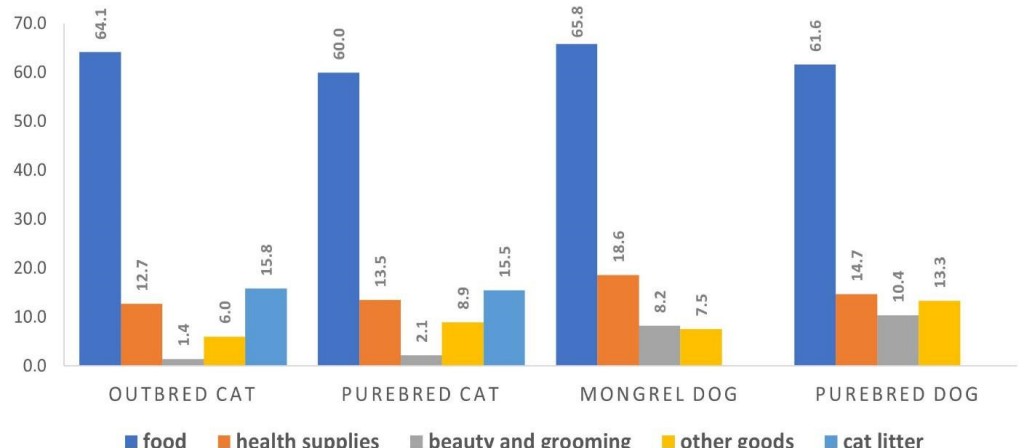

**Figure 5.** The cost breakdown for owning a pet in Moscow (%), calculated based on the respondents' mean costs for each type of expenses.

### 3.5. Perceived Availability of Green Infrastructure

The majority of respondents (29%) do not walk with their pets, which is reflected in the large number of cat owners. It is not common to walk with cats in Russia. A quarter of respondents (24.3%) walk in the park near their homes, and 23.7% in the neighborhood. Special playgrounds for walking with pets are chosen by 16.2% of respondents. The "other" option was chosen by 6.9%, and specified that they either live in a private house or the countryside (Russian "dacha"), and their animals walk there in the yard. There were also such variants as in the forest, by a pond, on the street, and on the balcony.

In total, the availability of green infrastructure does not significantly correlate with dog or cat ownership (U = 10,379.000; Z = −0.088; $p$ = 0.930). When respondents were asked to assess the availability of green infrastructure in their district, dog owners did not give more than 4 points to the administrative districts (Table S3 in Supplementary). The maximum score given by the cat owners was 4.2. There is statistical dependence according to the district where respondents live (chi-square = 34.080; df = 9; $p$ = 0.000). The most valuable district was stated as the Eastern Administrative District (mean = 4.05), and less valuable were the Central Administrative District (mean = 2.75) and the Southeastern Administrative District (mean = 2.77). These estimates correlate with data on the percentage of green areas: Eastern Administrative District (60.3%), Central Administrative District (23.5%), Southeastern Administrative District (37.4%).

## 4. Discussion

The study aimed to improve our understanding of the perceived benefits and costs of owning pets in a city from the ES perspective. To do so, we developed an approach to translate the perceived benefits and risks of owning a pet into ES and the disservices and NCP frameworks based on a social survey, and implemented it for the case of the Moscow megapolis. While there is a wealth of studies on values associated with owning a pet in the urban environment [25,45,46], to the best of our knowledge, this is the first attempt to fit them into ES classifications. We used CICES classification for ES and NCP to showcase how reasons for owning a pet translate into ecosystem services. In both, there are the classes of provisioning/material, regulating and cultural/non-material services from livestock and wild animals [38,39]; however, the benefits derived from pets such as dogs and cats are not explicitly named in either of them. Similar investigations into "domesticated" ecosystems and their representatives, such as ornamental plants [47] or home gardens [48], have revealed the many different values people derive from them, which are often overlooked when concentrating on either wild or more conventional production systems. On a more conceptual level, both ES and NCP have been criticized for being a too utilitarian, anthropocentric or dualistic representation of human–nature relationships (recent example—[49]), thus also suggesting that they provide an incomplete reflection of the benefits people derive from nature.

As our findings demonstrate, classes of ES and NCPs could be supplemented by explicit inclusion of the benefits derived from owning pets as natural biotic elements. Provisioning services (1.1.3.1 in CICES and 12 in NCP) are mostly relevant for residents of the countryside or people engaging in hunting, and for those areas where cats and dogs are still a source of nutrition for people. Regulating services represent pest control functions in the form of rodent control or safe-guarding people from other threats (2.2.3.1 in CICES and 10 in NCP). Cultural services are connected to direct physical activities such as walking or playing with dogs for the purpose of improving physical health (3.1.1.1 in CICES and 16 in NCP); communicating or interacting with pets for mental health or aesthetic appreciation (3.1.1.2 in CICES and 15 and 16 in NCP); direct intellectual interaction for educational values (3.1.2.2 in CICES and 15 in NCP), and indirect spiritual, symbolic interaction—the sense of place (3.2.1.1 in CICES and 17 in NCP). At the same time, some of the benefits obtained from owning a pet are not reflected in ES classifications, even implicitly. For example, in CICES, there is no class for social interaction through ecosystems or biota, whereas parks and urban green spaces are meeting places for people and pets, and dogs in particular can be a cause for people to meet or start interactions, and as a result might improve neighborhood social capital [50].

We find that while some of the reasons for owning pets statistically differ for dogs and cats among residents of Moscow, overall, pets are no longer valued solely for providing instrumental and practical functions (catching mice, guarding a home), but rather substantial shares of the reasons for owning a pet are linked to cultural services (93%). Indeed, today, in urban areas, cats and dogs are more the "suppliers" of entertainment, enjoyment, and emotional support, as has been observed by other studies (e.g., [1,10,11]). Dogs are recognized as a source of joy and social support that might assist owners in increasing their health and physical activity [13,14]. On the other hand, cat ownership is often associated with mental health and a sense of home. The benefits derived from companion animals include reductions in depression and loneliness, while enhancing social interaction or social skills [51]. The majority of values in relation to pets linked to the cultural ES (as opposed to historically more important provisioning/material ones) have also been detected for other ecosystems or their characteristics, for example, for the ES obtained from home gardens [48]. This is in line with the current call in research to recognize and bring forward the plurality of cultural ES (beyond recreation and aesthetic appreciation) and relational values, as more and more studies reveal their significant contributions to human well-being [52–55].

Apart from these various services, pet owners face a number of problems and disservices, such as allergies, damaged furniture and things, noise, etc. Nevertheless, these are not considered critical enough to prevent the respondents of our study from owning a pet. Humans do not perceive zoonoses and infectious diseases as a threat, since their pets have never infected them, and the process of acquiring zoonotic infections from the environment is often independent of domestic animals [56]. Interestingly, the impact of pets on wildlife is not recognized as a risk among our respondents. Only a few of them considered it as a problem that their pet kills birds and small animals, in contrast to countries (Australia, New Zealand, the UK, the USA, China, Japan, etc.) where the risks of extinction of wildlife, the direct negative impact on population size and dynamics [57], and the problem of predation by cats and dogs on native fauna [27,58,59] are acutely recognized in society. These risks and problems are widely discussed in terms of management measures and regulation policies. Thus, in New Zealand, public support for population control methods for unowned cats is being explored [30]. In Australia, scientists and society are searching for various management actions to control dog and cat populations and behavior [60].

Our results also indicate that people are ready to pay for pet ownership despite these potential risks and problems, and there is a significant difference between the costs for owning a dog and a cat ($p = 0.000$). This may be the reason why cats are the most common pets, both among the participants in our survey and according to other studies [36]. The market of the pet industry is large, and growing each year. Pet ownership has been valued as USD 1.44 billion, that is, 0.56% of Moscow's gross regional product. According to our study, the average annual cost per animal in Moscow reported by participants is around USD 519–816, that is, 3.5–5.3% of the average salary in Moscow [61]. In comparison, in Canada, the cost is around CAD 1000–1500 [33,34]. The median yearly veterinary expenditure in New Zealand is about USD 200–499 by dog owners and USD 100–199 by cat owners [62]. Moreover, maintenance costs have been growing significantly recently due to the arising of more specific types of care, such as beauticians, walking services, and pet entertainment.

This study has several limitations. First of all, the sample size is relatively low and is biased in relation to gender and completed education levels. The sample could have been biased due to the sampling method employed in this study, namely, the recruitment of participants using social media websites and targeting specific groups. The most common pitfall of such methods is the under-representation of the older generation. While our sample is slightly skewed towards people younger than 40, we also obtained answers from pet owners in older groups. Secondly, our study's design targets pet owners only, and does not reflect the perspective of those not owning a pet. This exclusion was intentional; however, our findings should only be applied in the context of pet owners in the urban environment, and not the urban population in general. Thirdly, data for this study were collected during a lockdown due to the COVID-19 pandemic outbreak. However, our design did not include questions to determine the potential effects of this on answers, and thus no conclusions about it can be drawn. Finally, we conducted this study on Russian nationals, and its findings should be cautiously extrapolated to other parts of the world. However, we believe that as a first attempt, they still provide a good first impression on the overview and extent of ES and disservices linked to owning a pet in a megapolis.

## 5. Implications for Urban Planning

More than half of the global population reside in urban environments, with further increases expected leading up to 2050 (UN 2010). Urban nature, including green and blue spaces as well as biodiversity, supports urban residents' well-being and prosperity [63]. ES can increase resilience and quality of life in cities through a number of ways [64]. The ES concept is also considered useful to support decision-making as well as spatial planning for both practitioners and scientists [65]. Cultural ES is among the most important types of ES in urban environments, providing means for the re-establishment and maintenance of connections with nature, which are even more important for the urban population [66].

Lockdowns and other restrictions introduced to combat the COVID-19 pandemic have further exacerbated the need to pay extra attention to any contributors to the mental and physical health of urban residents [67,68]. It is not a far-fetched conclusion that pets have been an important source of positive experiences and emotions (e.g., an example from Malaysia is discussed in [69]) and provide complimentary social support to supplement virtual human-to-human interaction [70]. At the same time, there have been reports of negative effects related to fears of pets transmitting disease [71], or even an association between delays in testing and entering treatment for COVID-19 among pet owners in the U.S. due to worries of who will take care of their pets [17]. Therefore, the findings of this study demonstrate that domestic animals such as dogs and cats contribute to urban dwellers' well-being (both positively and negatively), especially in relation to cultural services, making it beneficial and important for ES approaches to include such values explicitly (as services and disservices, respectively). Such explicit inclusion in classifications of ES and NCP would serve as a first step towards the evaluation of these services and ultimately consideration in decision-making and planning.

Decision-making processes in urban planning ought to focus more on benefits for residents [72]. In order to achieve sustainable futures, there is also a need to investigate and embrace a plurality of values, as well as differences between groups of residents/stakeholders [52,66]. As the number of urban pets grows, and the care industry expands, so does the distinct stakeholder group of pet owners, with their own specific needs and demands. The existence of such a group creates a need to address different questions connected to animal management policy, such as microchipping and pet registration [62], walking zones [45,73–75], impact on biodiversity [28,58,60], and the regulation of abandoned and wild animals in the urban environment [62]. The COVID-19 pandemic has also introduced more concerns regarding pets, such as (1) how to care for pets during such crises in the future, (2) whether there are zoonotic concerns associated with caring for a pet, and 3) what repercussions there are for pet care [69,76]. Thus, urban planning is challenged with addressing these and other demands of pet owners, while balancing them with those of other interest groups, for instance through the cooperation between multiple beneficiaries and the identification of stakeholder-specific multifunctionality hotspots [77].

## 6. Conclusions

Domestic animals such as cats and dogs contribute to the well-being of their owners in urban environments; however, this is not reflected in the existing ES and disservices classifications, and thus presents an obstacle to their assessment and inclusion into decision-making. In this study, we developed and tested an approach in which data on the reasons, disadvantages and costs of owning a pet in a megapolis, collected by means of a survey among residents/pet owners, are translated into the ES and disservices associated with pet ownership. Our findings show that the most appreciated ES group among Moscow-residing pet owners is cultural services linked to mental and physical health, aesthetic enjoyment, educational values, and others (93%), whereas regulating and provisioning services linked to practical and material benefits are less valuable for megapolis residents (4 and 1%, respectively). It is possible and promising to use the ES concept to analyze the interaction between humans, domestic animals, urban green areas and biodiversity for better urban planning and to facilitate sustainable well-being.

**Supplementary Materials:** The following are available online at https://www.mdpi.com/article/10.3390/su131910596/s1.

**Author Contributions:** Conceptualization, A.K. and V.M.; data curation, A.K.; formal analysis, A.K., V.M. and A.F.; investigation, A.K., V.M., A.F. and V.V.; methodology, A.K., V.M. and A.F.; resources, A.K., V.M., A.F. and V.V.; software, A.K.; supervision, A.K. and V.M.; validation, A.K. and A.F.; visualization, A.K. and V.M.; writing—original draft, A.K., V.M. and A.F.; writing—review and editing, A.K., A.F. and V.V. All authors have read and agreed to the published version of the manuscript.

**Funding:** The social survey and translation into the ecosystem services was supported by the Russian Science Foundation project under grant number 19-77-30012. Data processing and paper preparation were supported by the RUDN University Strategic Academic Leadership Program.

**Institutional Review Board Statement:** Not applicable.

**Informed Consent Statement:** Not applicable.

**Data Availability Statement:** The data presented in this study are available on request from the corresponding author. The data are not publicly available due to privacy restrictions related to anonymous participation in the survey.

**Conflicts of Interest:** The authors declare no conflict of interest. The funders had no role in the design of the study; in the collection, analyses, or interpretation of data; in the writing of the manuscript, or in the decision to publish the results.

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
