# Peer review of "Perceived Benefits and Costs of Owning a Pet in a Megapolis: An Ecosystem Services Perspective"

_sustainability, doi:10.3390/su131910596_

Round 1

Reviewer 1 Report

This paper reports results of an online survey in Moscow on perceived benefits and costs of pets, split into dogs and cats, and attempts to translate these into categories of the ecosystem services / disservices approach. The insights from the survey add to those of other studies on domesticated animals. The argument to consider pet related effects in classifications of ESS or disservices is a useful contribution to ongoing discussions.

My major concern is that differentiating results for cats and dogs is an important focus of the study. Yet these differences are only being described without statistically testing for significance. These additional analyses should be certainly performed with adequate methods and the results reported in detail.

The structure of the paper should better reflect its two components: the survey and the discussion on ESS categories. While much place is used to report and discuss details from the survey, the introduction and discussion tends to be biased by the ESS perspective.

Other points

The title is too long, hard to understand and does not adequately reflect the approach of the manuscript. From my understanding, the main focus is on perceived benefits and costs of pets in Moscow and how these could be related to categories of ESS and disservices.  Implications for urban planning are addressed in the latter part of the MS, but are not so strong as to be reflected in the title.

Abstract: The beginning with well-known insights should become shorter and the justification for performing the survey in Moscow and for relating the results to ESS etc. should be clarified. (You could discuss the pet/ESS-relationship well based on the existing literature). Would be good to reflect your approach by at least two research questions.   

Please rephrase this sentence:   “We advocate that in urban areas pet owners are an essential and increasing in numbers stakeholder groups, interests and needs of which ought to be taken into consideration in planning and management”

The graphical abstract is very nice.

Line 59: Indicate geographical context. Otherwise it would hold for Asia.

Lines 78-79: This statement needs a reference.

Lines 86-87: Hall et al. is no ecological study. See Buchholz et al. 2021 as an urban study (and references therein): https://www.sciencedirect.com/science/article/abs/pii/S016920462100164X

and Doherty et al. as an non-urban study: https://www.sciencedirect.com/science/article/abs/pii/S0006320717305967

Line 106: Please address disservices as well.

The introduction should end with detailed research questions that will guide the following analyses.

Line 121-25: These are assumption that need to be referenced or better deleted here as they do not describe the study area.

It would be nice to document an English version of the survey in the supplemental materials so that others can replicate the study.

As there are a range of similar studies, you should relate your approach to previous work and justify the selection of parameters according. You might choose the same parameters to compare results from different regions – or incorporate new parameters to address knowledge gaps.

Line 169: not necessary to mention: “Diagrams were created in Excel”

Tab. 1: Could you provide corresponding data for the Moscow population?

Line 185: Did they prefer or own the respective types of dogs?

Fig.2: Cultural ESS should be indicated in this figure as you refer to them in the text.

Discussion: relating results of your survey to the existing literature could be strengthened.

Line 334: Needs to be referenced. Wood et al. e.g., is on social interactions: https://www.ncbi.nlm.nih.gov/pmc/articles/PMC5769067/

Line 336: I understand what you mean with practical reasons, but this term is ambiguous here. Perhaps merge with the following sentence.

Line 354-55: This is a circular argument as you only surveyed people owning a pet. That your study did not compare pet owners and non-owners (as other studies did) is a flaw of your approach which you should discuss.

Line 371: Silva et al. refer to native species!

Line 383-84: This sentence is not correct since there is a wealth on studies on “values associated with owning pets in urban environments”.  Not necessary to repeat the approach of the study here. Just address the limitations directly. There are much more limitations as addressed here. The sample size is relatively small, and the sample seems to be biased also towards gender and education. Importantly, you did not sample non-owners which strongly limits generalizations for the urban population. Please, carefully relate your results only to the group of pet owners.

Line 399: two sentences?

Start a new sentence in Line 406

Line 406-07: Googe Scholar reveals a range of recent publications on the role of pets during the Covid pandemics which should be considered.

The conclusions section should become shorter, without repeating too much results.

Reviewer 2 Report

This is a very interesting and neatly conducted work on the domestic animals' contribution to the ecosystem services framework. The Authors argue that the pets services to people should be recognized as part of ecosystem cultural services, the concept is interesting but it seems that the idea of pets being a part of the natural ecosystem should be better supported (how can we justify pets to be part of the natural ecosystem? are megapolis' pets create some sort of collective ecosystem?). 

It would be recommended to include some reference to the covid-19 pandemic and lockdowns in the introduction section, that it became more prevalent in urban populations to adopt a pet. In fact, in many places, animal shelters got emptied. Also, reference to mental health therapy and self-care recommendation  - one of them that is very popular is to adopt a pet. 

The data collection was in the spring of 2020, wherein many countries the covid-19 lockdowns were mandated. Authors should discuss whether this could influence the responses or indicate it as a limitation. 

Since the costs of owning a pet are discussed in the context of a household, it should also be discussed in the context of a city. For example - the cost of cleaning and maintenance of the dirty dog owners, damage of the plants, cost of animal shelters and irresponsible owners -> abandoned pets. 

Reviewer 3 Report

The present paper is interesting. I really enjoyed reading and I appreciate the idea that the authors had. However, the authors did an enormous job in collecting data which should have been supported on a better statistical treatment. Still, I feel the paper is worth being considered, but specific matter must be at least considered.

Abstract needs to be reduced by 65 words.

I missed numerical results. This means, when the authors explain that the article suggests or shows some finding, it needs to be justified, significance of the results or evidences of this findings being suggested.

I particularly enjoyed the graphical abstract.

Introduction

Please use citation formatting appropriately according to the journal guidelines.

Introduction is conducted appropriately but very long. I feel the same could be said in half the space. Please consider shortening and better defining paragraphs.

In my opinion, section 2.1. from material and methods is introduction. The only part I would keep as M&M is the three lines at the bottom (127 to 129).

Line 146. Please avoid using contraction as in didn’t.

I think a table explaining which inclusion and exclusion criteria were exactly would make this information rather clear.

I would split 2.2. section. Like in a first section dealing with how participants were contacted and selected. After this, what the human side sample of the study was in a separate section. Then deal with the questionnaire. At the end, I would separate stats in a single section.

One of my greatest concerns is…ok, the study is descriptive, but no test to compare frequencies is made, although authors do compare and state frequency differencies. Two alternatives come to my mind.

Two Proportion Z-Tests, which is a parametric alternative. Can be used if sample is larger than 30. Otherwise, t-test need to be used (if data is normally distributed). Observations need to be independent. Must be normally distributed. Data must have been randomly extracted from a population, which every element in the population having the same probability of being selected. When samples are compared, their sizes must be equal if possible.

If the aforementioned is not met, then a nonparametric approach should be used. Pearson’s Chi squared is the test to use.

The conclusions that this etst leads to are identical to those reached in the aforementioned z test, except for the fact that with z we estimate normal standard deviation (z). Then the square of z is identical to Pearson’s chi squared. Indeed they are calculated in the same manner by SPSS, and then you square root chi square statistic to calculate z

If samples are not independent then and single observations and variables are dicotomic, the option is McNemar test.

If samples are paired and observations repeated, Q Cochtrand test.

Discussion seems to be appropriately conducted, btu as I said I would have supported it on factual results. Otherwise, these results cannot lead to conclusions, they are just observative.

Conclusions need to be shortened, but I liked their content.

Round 2

Reviewer 1 Report

The authors have presented a very good revision! The additional statistical analyses make the text even more meaningful. In my view, the MS is now ready for acceptation. 
I have only very minor comments:
Please check if it is not more useful to change the order of the last two sentences in the introduction.
In line 190 and 203 it is not necessary to indicate the significance level separately, because this is also apparent from the p-values.
Line 232: well-being in singular?
Please check 2). in line 348
"(e.g. example from Spain" in line 416 is awkward and should be reworded.

Reviewer 3 Report

I have no further comments

Author Response

Point 1. I have no further comments

Response 1: Thank you!